# Microarray-Based Prediction of Polycythemia after Exposure to High Altitudes

**DOI:** 10.3390/genes13071193

**Published:** 2022-07-02

**Authors:** Haijing Wang, Daoxin Liu, Pengfei Song, Feng Jiang, Tongzuo Zhang

**Affiliations:** 1Key Laboratory of Adaptation and Evolution of Plateau Biota, Northwest Institute of Plateau Biology, Chinese Academy of Sciences, Xining 810001, China; wanghj@nwipb.cas.cn (H.W.); liudx@qhu.edu.cn (D.L.); pfsong@nwipb.cas.cn (P.S.); jiangfeng@nwipb.cas.cn (F.J.); 2University of Chinese Academy of Sciences, Beijing 100049, China; 3Medical School, Qinghai University, Xining 810016, China; 4Qinghai Provincial Key Laboratory of Animal Ecological Genomics, Xining 810008, China; 5Kunlun College, Qinghai University, Xining 810016, China

**Keywords:** high altitude polycythemia, logistic model, prediction, microarray

## Abstract

In high-altitude environments, the prevalence of high-altitude polycythemia (HAPC) ranges between 5 and 18 percent. However, there is currently no effective treatment for this condition. Therefore, disease prevention has emerged as a critical strategy against this disease. Here, we looked into the microarray profiles of GSE135109 and GSE29977, linked to either short- or long-term exposure to the Qinghai Tibet Plateau (QTP). The results revealed inhibition in the adaptive immune response during 30 days of exposure to QTP. Following a gene set enrichment analysis (GSEA) discovered that genes associated with HAPC were enriched in Cluster1, which showed a dramatic upregulation on the third day after arriving at the QTP. We then used GeneLogit to construct a logistic prediction model, which allowed us to identify 50 genes that classify HAPC patients. In these genes, *LRRC18* and *HCAR3* were also significantly altered following early QTP exposure, suggesting that they may serve as hub genes for HAPC development. The in-depth study of a combination of the datasets of transcriptomic changes during exposure to a high altitude and whether diseases occur after long-term exposure in Hans can give us some inspiration about genes associated with HAPC development during adaption to high altitudes.

## 1. Introduction

With their hypoxic circumstances, high altitudes have a profound impact on the physical health of locals and visitors [1]. In order to adapt to hypobaric hypoxia, dwellers from plain to plateau must allow the lungs to acquire more atmospheric oxygen, and the number of red blood cells (RBCs) will rise compensatively [2,3]. Mild polycythemia is considered a beneficial response to high-altitude hypoxia [4]. However, in certain people, the number of RBCs continues to grow, resulting in high altitude polycythemia (HAPC), which is characterized by headache, disorientation, sleeplessness, and bone discomfort. It was first described by Viault in 1980 [5] and is the most typical characteristic sign of chronic mountain sickness [6].

Over 140 million people reside at altitudes greater than 2500 meters above sea level [7]. The Qinghai-Tibet Plateau (QTP), the highest and largest plateau in the world, is characterized by low oxygen levels, and millions of people live and work in this region. According to reports, the prevalence of HAPC in the QTP increases with altitude and ranges between 5% and 18% [8,9]. As the Qinghai-Tibet Railway was finished, an increasing number of Han people migrated and traveled to Tibet. HAPC is much more prevalent among Han than among natives [2]. Historically, studies focused more on the pathophysiologic processes of HAPC [10]. Furthermore, numerous investigations have verified the associations between gene polymorphisms and HAPC by analyzing the genetic backgrounds of Han people and Tibetans [9,11,12]. Nonetheless, disease-related genes require further investigation, particularly in Han people. Here, we explored the microarray profile of HAPC and discovered that genes involved in the development of this disease even changed at the early time of exposure to hypoxia.

This study investigated the microarray profiles of GSE135109 [13] and GSE29977 [6], respectively. We classified the differential expression genes (DEGs) in GSE135109 into 6 clusters with different dynamic expression patterns. Furthermore, 415 DEGs in GSE29977 were used to construct a logistic model to predict HAPC. At last, we overlapped the genes identified in two datasets and acquired *LRRC18* and *HCAR3* as hub genes related to HAPC development after exposure to QTP. This research will give us some inspiration about key genes associated with HAPC development during adaption to high altitudes and significantly enhance the early warning for HAPC.

## 2. Materials and Methods

### 2.1. Data Collection and Preliminary Processing

We searched in NCBI-GEO (https://www.ncbi.nlm.nih.gov/geo/, accessed on 17 March 2022), using the keywords “high altitude” and “Homo sapiens” and downloaded the microarray datasets regarding people adjusting to high altitudes, including GSE135109 [13] and GSE29977 [6]. The GSE135109 microarray dataset comprised 8 samples, which were sampled at a low-altitude starting point (500 m) and at 3, 7, and 30 days after a quick ascent to a high altitude (5200 m). The volunteers in this dataset were never previously at high altitudes and had not developed HAPC during 30 days of exposure to high altitudes. Using the GPL24539 platform, white blood cells from the periphery were collected and sequenced. The oligo package was used to read CEL files from the GSE135109 dataset by annotating gene symbols. The GSE29977 microarray data included 10 subjects who were Hans and had migrated to the high altitude (4550 m) for more than 8 months, 5 of which were high altitude polycythemia (HAPC) patients, while the remaining 5 were matched controls. The peripheral blood leukocytes of 10 individuals were collected, and the gene expression profiling was analyzed by GPL570 platform. GEOquery and AnnoProbe packages were conducted to download and annotate this dataset. The participants in both two datasets were young men. After the collected datasets were annotated with official gene symbols and normalized (log2), we acquired 18,190 genes and 18 samples for the following analysis.

### 2.2. Identification of Differential Expression Genes

A differential expression gene (DEG) analysis was performed on the GSE135109 and GSE29977 datasets, respectively. The limma (v 3.50.0) package [14] and the normalizeBetweenArrays function were used to normalize the quantiles. The significant DEG threshold was defined as log(FC) with an absolute value more than 1 and a *p*-value less than 0.05.

### 2.3. Microarray Time-Course Analysis

The union DEG sets (1135 genes) between three groups (D03vsD0, D07vsD0, D30vsD0) were acquired for a further time-course analysis. The avereps function in the limma package was used to compute the average expression values at various times. The Mfuzz package [15] was implemented to detect clusters with consistent expression trends during exposure to QTP.

### 2.4. Logistic Prediction Model Using Procedure Logistic

To develop a logistic prediction model, the GeneLogit software, a logistic regression method applied to microarray data in small samples, was utilized [16]. Firstly, the localFDR function was used to estimate the false discovery rate (FDR). Secondly, the theta estimate provided by the model estimation function was 0.012. Thirdly, the bootstrap prediction function was used to find the optimal τ and the corresponding prediction error for different values of q. The results are displayed in Table 1. Finally, we selected the procedure logistic (q = 50, tau = 0.379) to construct our prediction model.

### 2.5. Analysis of Weighted Gene Co-Expression Network Analysis (WGCNA)

We conducted the weighted gene co-expression network analysis (WGCNA) to discover the relationships between gene expression patterns and phenotypes in GSE29977 using the WGCNA package [17]. For the subsequent analysis, we kept 14043 genes with a median absolute deviation (MAD) greater than 0.01. The soft power was estimated at 14, and we constructed an unsigned scale-free co-expression network for the genes with a minModuleSize of 100 and mergeCutHeight of 0.25. Ultimately, we identified 35 modules. The moduleEigengenes function calculated the dissimilarity of the module eigengenes (MEs), and then the Spearman correlation was used to assess the associations between the MEs and phenotypes. The exportNetworkToCytoscape function was used to export the network of genes selected in the logistic prediction model. At last, Cytoscape software [18] was acquired to visualize the network.

### 2.6. Evaluation of Immune Cell Types

The CIBERSORT algorithm was applied to the log transformed normalized microarray data of GSE135109, and the leukocyte signature matrix (Cibersort: LM22) was used to infer immune cell types in each sample [19]. We utilized 1000 permutations from the default signature matrix and calculated the *p*-value and root mean square error for each sample file to improve the accuracy of the deconvolution algorithm. The scores for each signature were summarized for each group and median centered to permit comparisons between groups. The wilcox.test was used to discover the significance between comparisons.

### 2.7. Functional and Pathway Enrichment Analysis

A Gene Ontology (GO) and Kyoto Encyclopedia of Genes and Genomes (KEGG) pathway enrichment analysis and visualization were performed using the package clusterProfiler [20] for genes with different expression patterns. The GO comparison across clusters was performed using the compareCluster function. The gene set enrichment analysis (GSEA) [21] was performed to determine whether a series of pre-defined geneset were enriched in the gene rank. The Molecular Signatures Database (MSigDB) [22] was also used here. *p*-values of <0.05 denoted statistical significance.

## 3. Results

### 3.1. Exposure to the QTP Weakened the Adaptive Immune Response in Leukocytes

To determine the differences in the gene expressions of the four groups, the principal component analysis (PCA) was conducted (Figure 1A). The apparent distinction was that the gene expression changes after exposure to the QTP for 3 days were similar to those for 7 days (increasing along the PC1 axis). However, this trend reduced along the PC2 axis after 30 days, indicating that the continuous exposure altered the expression of genes in white blood cells. Then, the differential expression gene analysis was conducted to identify the differential expression genes (DEGs) between different time points (D03, D07, D30) and the control group (D0). The Venn diagram demonstrated that 108 genes were consistently significant along with the exposure duration (Figure 1B). We performed GO enrichment to determine the function of these 108 genes, and the dot plot revealed that these genes were enriched in the adaptive immune response category (Figure 1C). We extracted 12 genes related to the adaptive immune response, and the log2 fold change of each treatment group declined with exposure time (Figure 1D). These findings suggest that hypoxia in the QTP might inhibit the adaptive immune response of white blood cells but that it would recover as the exposure duration increases.

### 3.2. Dynamic Expression and Functional Characteristics of Leukocytes upon Arrival in the QTP

We investigated the gene expression dynamics in human leukocytes at four time points using 1135 genes that were significantly different from the control, and six clusters were obtained (Figure 2A). Cluster 1 (206 genes) and cluster 3 (244 genes) were dramatically changed with an increase in cluster1 and a decrease in cluster 3 on the third day after arrival in the QTP. The dynamic expression of these 450 genes exhibited the greatest altitude sensitivity. Then, we performed an enrichment analysis to determine their function (Figure 2B). Both overlaps and variances existed between the two clusters. Cluster 1 was specifically enriched in terms of the intrinsic apoptotic signaling pathway and the cellular response to interleukin-1, and *HIF1*α was also found in this cluster. Cluster 3 was enriched in terms of receptor-mediated endocytosis and adaptive thermogenesis. Both of them were enriched in the positive regulation of cytokine production.

Cluster 2, cluster 5, and cluster 6 had distinct expression patterns, and the enrichment of these clusters revealed no relationship between them (Figure 3A). The genes in cluster 2 increased with the time of exposure and were enriched in the positive regulation of hemopoiesis and related to leukocytes. In contrast, genes in cluster6 decreased and were enriched in the regulation of lymphocytes. The genes in cluster5 showed a stable state after three days of arriving at the QTP and were enriched in oxygen transport. Leukocytes consist of different immune cell types, such as lymphocytes, monocytes, and neutrophils. To discover the immune cell types in samples, we conducted CIBERSORT algorithm. In this case, 17 types were acquired, none of which displayed the same expression profile (Figure 3B and Appendix A). This result reminded us that the expression patterns in leukocytes were in entirely different forms.

### 3.3. Gene Expression in Leukocytes of HAPC Patients Exhibited Substantial Individual Variation

The GSE29977 dataset contains the microarray expression levels of 5 patients and 5 healthy human blood leukocytes that are paired with additional personal information. However, the PCA revealed significant individual differences, indicating a large dispersion in the group (Figure 4A). Next, we performed DEG analysis and acquired 212 downregulated and 203 upregulated genes (Figure 4B). Then, the gene set enrichment analysis (GSEA) was conducted to figure out the potential function, and the result indicated that genes involved in the metabolism of heme were downregulated and HBZ was included in this geneset (Appendix A).

To figure out the relationship with DEGs in GSE135109 and GSE29977, we used the GSEA. The result was only enriched in cluster1 (Figure 4C), which increased quickly upon QTP exposure and also reminded us that genes related to HAPC were affected at an early stage of exposure to high altitudes. However, when we examined genes that were overlapped in two datasets, a few of them were identified (Appendix A). In order to figure out the molecular mechanism underlying HAPC, we obtained the pivotal modules from the WGCNA analysis (Figure 4D and Appendix A). The steelblue and salmon modules were significant in the HAPC patients. The blue module was also negatively correlated with Qinghai CMS scores but positive with oxygen saturation. This meant that these three modules were directly related to the disease. The oxygen saturation was also relevant with the white, royalblue, and purple modules. Additionally, the length of time on the plateau showed different signs and was significant with modules of grey60, datkturquoise, tan, and saddlebrown. This result demonstrated that the age, time on plateau, and oxygen saturation also impact the disease outcome. The steelblue module drew our attention. It was not only related to the disease phenotype, but also to CMS which is its diagnostic criterion (Figure 4D). Therefore, we further analyzed the genes in this module. The results showed that the biological process of this gene set was enriched in the Wnt signaling pathway (Figure 5A), and the homeostasis in leukocytes was inhibited by genes in this module (Figure 5B).

### 3.4. Construction of a Prediction Model for HAPC

To identify the marker genes that predict HAPC, we constructed the logistic regression model using the 415 DEGs. The optimal *τ* and the corresponding prediction errors for *q* = 1, 10, 20, 50, and 100 are given in Table 1. The optimal *τ* decreases as *q* increases and choosing a *q* value beyond 50 does not help much in further reducing the prediction error. Next, we chose the procedure logistic (*q* = 50, *τ* = 0.379) to build our prediction model, with the results given in Table 2, and obtained 50 marker genes to construct the prediction model. However, there was no separate test dataset to assess the prediction error for our model. Then, we used this dataset itself to cross-validate the model and illustrated a higher efficiency (Appendix A). *LRRC18* and *HCAR3* were also overlapped in the DEGs of GSE135109 (Appendix A). To visualize the contribution of these genes, we built a network of these 50 marker genes using a weight of edge generated threshold of 0.02 in WGCNA (Figure 6). *HCAR3* showed higher total and inner connectivity (kTotal) in this network. Here, we constructed a logistic model to screen the marker genes to predict HAPC.

## 4. Discussion

After exposure to the plateau environment, the changes in hematopoiesis are more complicated in humans than in experimental animals. This study merged two datasets regarding the expression levels of leukocytes in peripheral blood in humans exposed to QTP and identified genes that could predict HAPC. The results indicated that the acclimation to high altitude affects the immune system and that immune suppression would be restored as time progressed. HAPC patients and the adapted people had distinct expression patterns, and the steelblue module was related to both the disease phenotype and CMS. Then the prediction model for HAPC was also developed, and significant alterations in *HCAR3* and *LRRC18* were also observed during the beginning of the exposure to a high-altitude environment, suggesting that HAPC patients can be identified early.

At short timescales, the impact on the adaptive immune response cut a striking figure. The adaptive immune system relies on B cells and T cells, and in vitro experiments demonstrated that the human adaptive immune response is dependent on the oxygen level [23]. CX3CR1 is stably expressed on CD8+ T cells [24], and its expression was highly suppressed on the third day following a quick climb to a high altitude. Physiological hypoxia impacts the immune cell function, ultimately controlling innate and adaptive immune responses, mainly though transcriptional regulation via hypoxia-inducible factors (HIFs) [25]. Here, we showed that HIF1α was rapidly upregulated and associated with multiple biological processes, including intrinsic apoptotic signaling. Severe acute hypoxia promotes oxidative damage, which could trigger apoptotic mechanisms [26]. Nonetheless, a study conducted on the Kyrgyz population revealed that circulating levels of the apoptotic signal were diminished following acute and chronic exposure to high altitude conditions [27].

Inflammatory factors are reduced with acclimations to high altitude and are closely related to the pathogenesis of HAPC and high-altitude acclimation [28]. During the first month of exposure, immune-related genes were mainly affected. Genes associated with lymphocytes were decreased in leukocytes and increased in lymphocytes. The CIBERSORT predictions also suggested that the expression profile of white blood cells exhibited inconsistencies. This result committed to the study at 3700 m demonstrated an increase in neutrophils and a slight decrease in lymphocytes [29] and exposure to acute hypoxia marginally raised the neutrophils [30]. However, several investigations revealed contradictory results. Hypoxic preconditioning combined with altitude training increased T lymphocyte CD4/CD8 expression [31]. The latest cytological evidence showed that metabolic stress under hypoxia rapidly drives T cell exhaustion [32]. Different immune cell types might be related to distinct metabolic features.

HAPC is an adverse consequence of the adaption to high altitude, whereby the RBC count continues to rise. As a result, significantly higher concentrations of hemoglobin were found in HAPC patients than in matched controls [6]. However, genes involved in the metabolism of heme were decreased. Heme is essential in detecting and utilizing molecular oxygen and must be synthesized and degraded within an individual nucleated cell [33]. A chronic hypoxia increased erythrocyte lifespan was discovered in a recent study [34], which might explain why the heme metabolism was inhibited in HAPC. Furthermore, the DEGs between HAPC and control subjects were also enriched in cluster 1, rapidly upregulated after high altitude exposure, and inhibited in HAPC patients. These results indicate that people who developed HAPC were sensitive to hypoxia. In this study, we discovered that the critical module related to the disease was enriched in Wnt signaling, and it was also found to be related to adaptation to high altitude in Ethiopia [35]. In addition, the Wnt pathway impacts the ability of Drosophila melanogaster to complete its life cycle under hypoxia [36], reminding us that Wnt signaling might serve an essential role in hypoxia adaptation and may disturb the steady-state in HAPC patients.

Prediction models are designed to assist individuals in making decisions on the use of diagnostic testing and starting or stopping treatments [37]. Here, we constructed a logistic prediction model to discover 50 genes to predict HAPC. Serum inflammatory factor profiles revealed several proinflammatory factors that were higher expressed, including IL-1 β, IL-2, IL-3, TNF-α, MCP-1, and IL-16 between the HAPC and plateau control groups in Han people [28]. However, the ages between the two compared groups showed a large difference (37.4 ± 9.7 in HAPC and 27.2 ± 3.5 in control), and aging may be associated with an upregulated inflammatory response [38]. In another study, plasma biomarkers of HAPC in older Tibetans (about 50 years old) were identified, including C4A, C6, CALR, MASP1, and CNDP1 [39]. Among these proteins, *MASP1* showed higher transcriptional expression (log_2_(FC) = 1.04, *p* value = 0.07) than others in this dataset. This inconsistency might remind us that the mechanisms between Tibetans and Hans in developing HAPC was differ.

*HCAR3* and *LRRC18* were also sensitive to exposure to high altitudes and might be able to predict whether a person would develop HAPC. *HCAR3* activates Gi signaling in immune cells and is also a therapeutic target for breast cancer and inflammatory bowel diseases [40,41,42], and these diseases cause hypoxia response. HCAR3 activates ERK1/2, which in turn activates NFκB-mediated hypoxia responses. Its transcriptional expression also is more downregulated in smokers than nonsmokers at altitude, which might be related to the inhibition of the hypoxic response [43]. *LRRC18* was also identified as a critical gene in coronary artery disease diagnosis [44], and its single-nucleotide polymorphisms are associated with systemic lupus erythematosus [45]. However, the relationship between these genes and the HAPC remains unclear. To investigate the underlying mechanism, additional research is required.

This study had several limitations. It was necessary to consider the restricted sample size while evaluating the results. Although split-sample validation was not performed, the goodness of fit of our logistic models was assessed. A cohort will be needed to evaluate the genes we identified here, along with a depth study of the mechanism of HAPC development.

## 5. Conclusions

To discover the molecular mechanism during human exposure to high altitudes, we explored two datasets related to short and long-term exposure to the QTP. The short-term microarray expression revealed an impaired adaptive immune response, and several immune cell types revealed dynamic alterations. In HAPC patients, the gene set for the heme metabolism was downregulated and also enriched in cluster1, which was immediately upregulated after arriving at high altitude. *HCAR3* and *LRRC18* expression might relate to the development of HAPC. Here, we established a model and preliminarily explored people who might develop HAPC.

## Figures and Tables

**Figure 1 genes-13-01193-f001:**
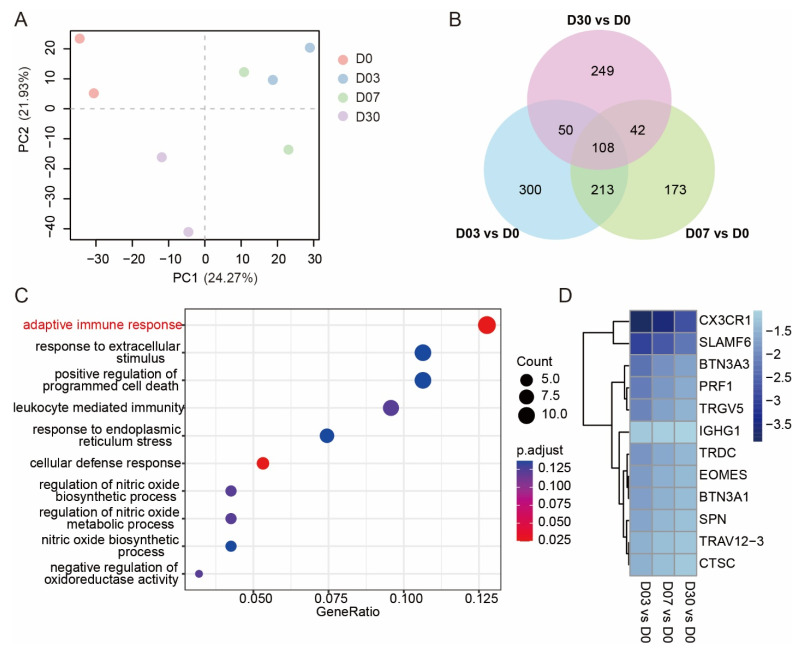
Altitude exposure weakened the adaptive immune response: (**A**) PCA plot of GSE135109; (**B**) Venn diagram showed the number of shared DEGs in three contrasts; (**C**) the dot plot was used for visualization of GO enrichment; (**D**) the heatmap showed log(FC) values in three contrasts of genes related to adaptive immune response.

**Figure 2 genes-13-01193-f002:**
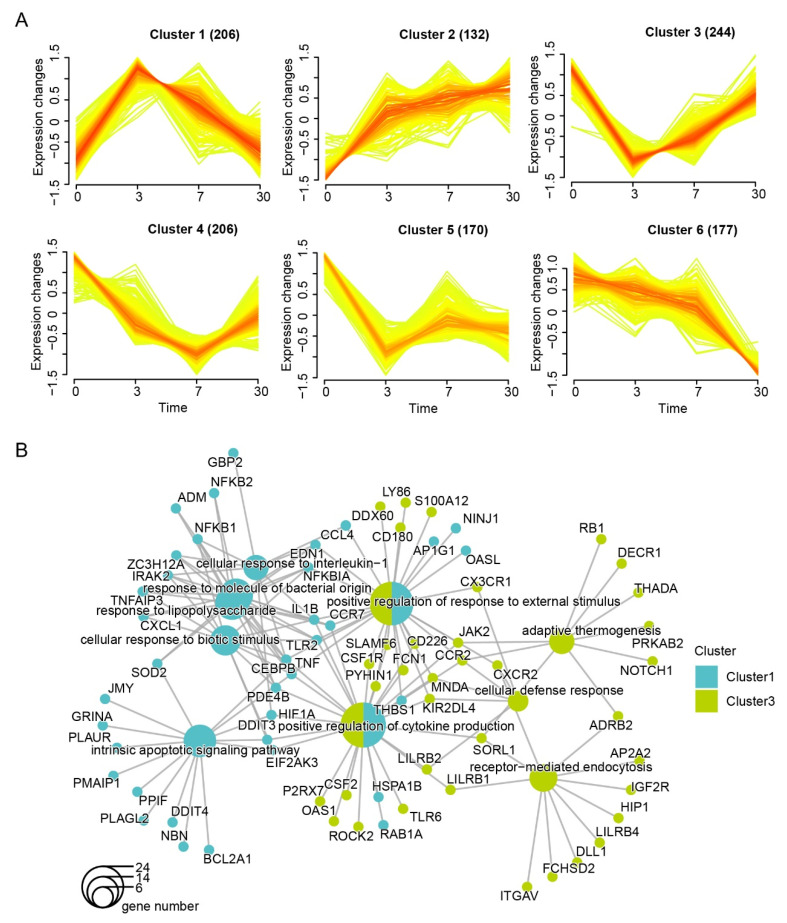
Dynamic changes in the expression of leukocytes: (**A**) time course plots of normalized data showing 6 patterns of gene expression of 1135 DEGs; (**B**) genes in cluster 1 and cluster 3 were compared to identify enriched biological processes, with the sizes of the dots representing the numbers of genes involved and with the colors identifying different clusters.

**Figure 3 genes-13-01193-f003:**
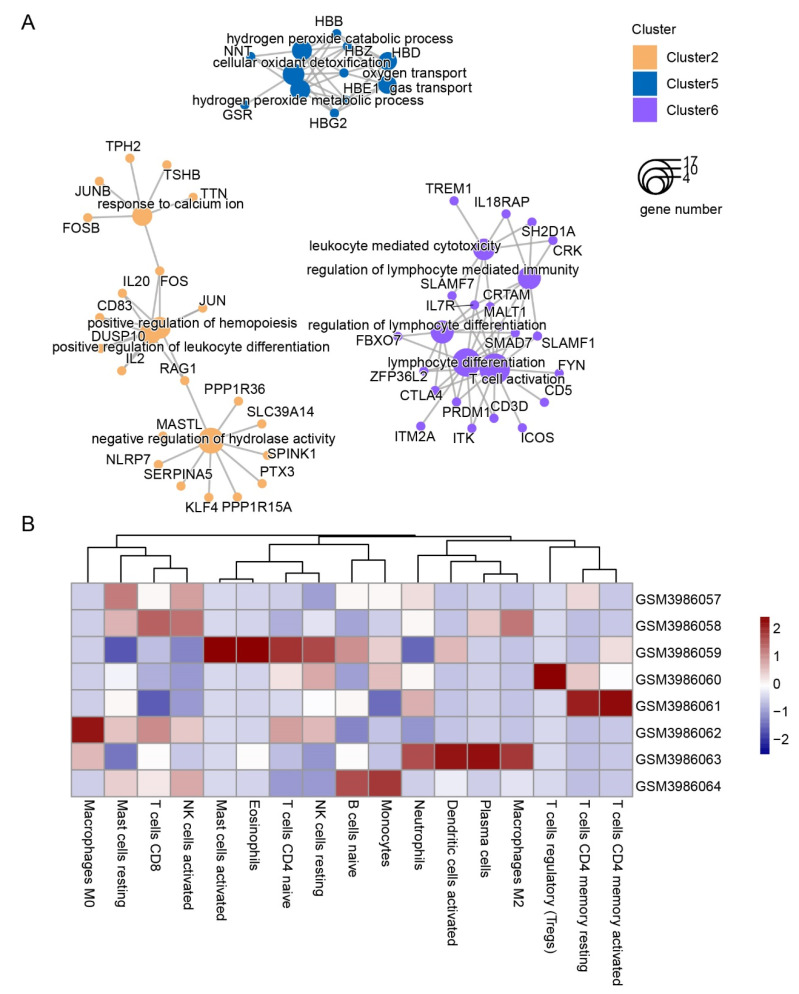
The biological functions of genes insensitive to hypoxia and the changes in prediction in immune cell types: (**A**) genes in cluster 2, cluster 5, and cluster 6 were compared to identify enriched biological processes, with the sizes of the dots representing the numbers of genes involved and with colors identifying different clusters; (**B**) the heatmap of the scores for each immune cell type.

**Figure 4 genes-13-01193-f004:**
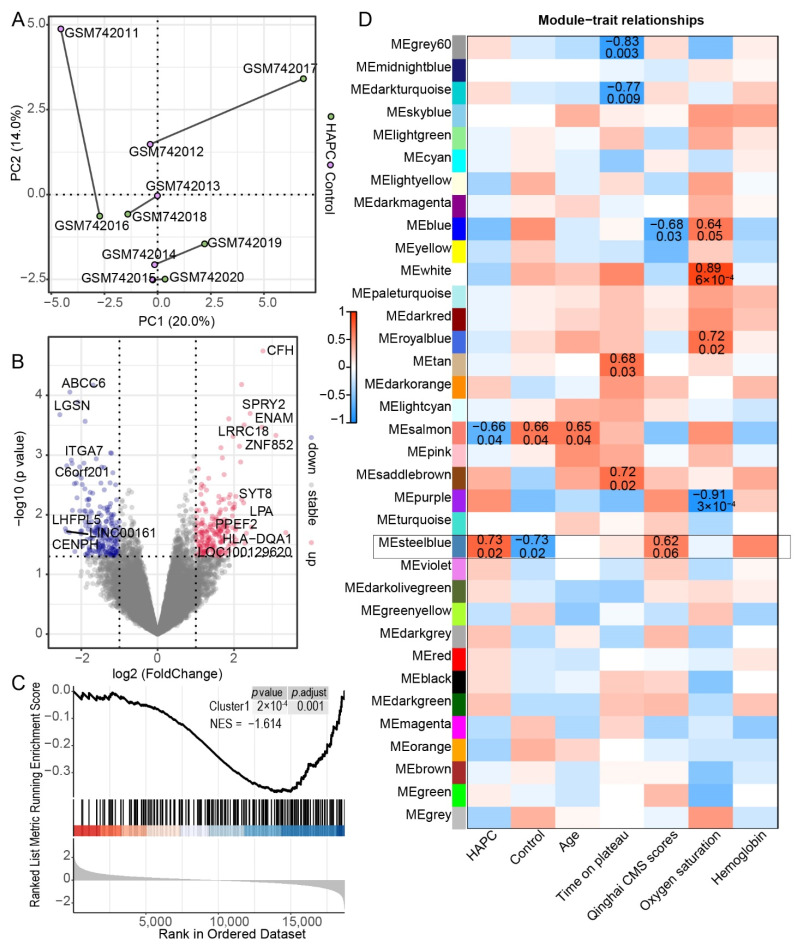
Analysis of genes related to HAPC in GSE135109: (**A**) PCA plot of GSE135109; (**B**) a volcanic map showing DEGs, where the red dots indicates DEGs with upregulation and the blue dots indicate downregulation; the genes with absolute log(FC) values higher than 2.25 were labeled; (**C**) the GSEA plot of geneset of cluster 1 in Figure 2A; (**D**) The heatmap of module trait relationships showing the correlations between each gene module and the phenotypes.

**Figure 5 genes-13-01193-f005:**
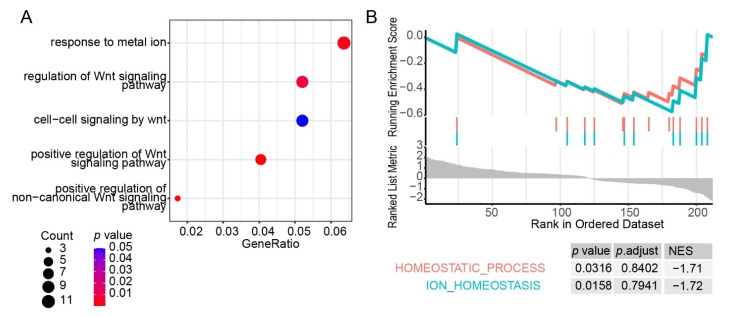
Enrichment of genes in the steelblue module: (**A**) the dot plot of functional annotations of genes in the steelblue module using the Gene Ontology biological process; (**B**) the GSEA plot of the “Homeostatic process” and “ion homeostasis” genesets in the steelblue module.

**Figure 6 genes-13-01193-f006:**
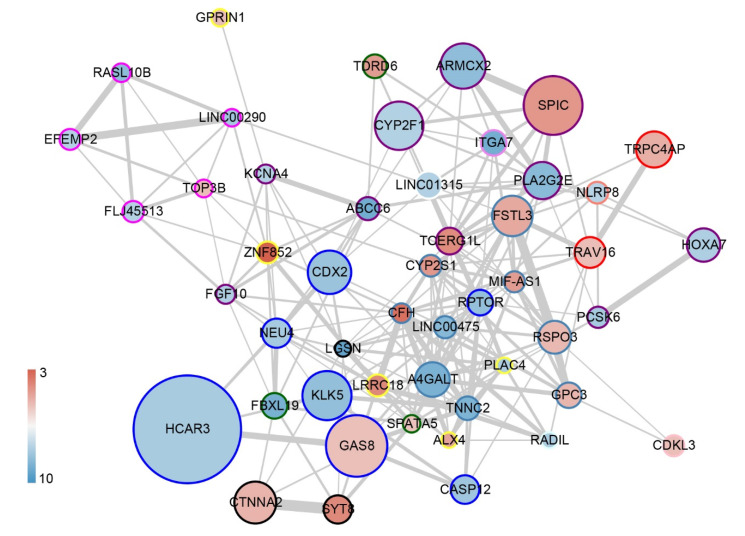
The co-expression network of marker genes predicting HAPC. The co-expression network visualization for marker genes was obtained from the WGCNA with a weight cutoff of 0.2. The colors of the dots indicate the log(FC) values, were red means upregulated genes and blue means downregulated. The colored circles of the dots indicate the modules to which they belong.

**Table 1 genes-13-01193-t001:** The optimal *τ* and the prediction error for different *q*.

*q*	1	10	20	50	100
Optimal *τ*	4.502	2.316	0.559	0.379	0.214
prediction error	0.059	0.058	0.056	0.051	0.051

**Table 2 genes-13-01193-t002:** Logistic prediction model using logistic regression (*q* = 50, *τ* = 0.379).

Gene	Intercept	*PLA2G2E*	*TNNC2*	*GAS8*	*RPTOR*	*GPC3*
coefficient	3.405	−0.078	−0.081	0.041	−0.067	0.056
Gene	*CFH*	*ZNF852*	*NLRP8*	*TCERG1L*	*FLJ45513*	*ARMCX2*
coefficient	0.107	0.122	−0.047	0.086	−0.042	−0.068
Gene	*MIF-AS1*	*A4GALT*	*ABCC6*	*LGSN*	*RADIL*	*NEU4*
coefficient	0.073	−0.081	−0.096	−0.105	−0.055	−0.052
Gene	*LINC00290*	*PCSK6*	*FGF10*	*SPIC*	*LRRC18*	*EFEMP2*
coefficient	−0.057	−0.064	−0.048	0.067	0.086	−0.048
Gene	*LINC01315*	*CYP2F1*	*CDX2*	*RASL10B*	*KLK5*	*RSPO3*
coefficient	−0.045	−0.044	−0.059	−0.070	−0.057	0.050
Gene	*PLAC4*	*TDRD6*	*ITGA7*	*KCNA4*	*SPATA5*	*HCAR3*
coefficient	−0.054	0.073	−0.082	−0.053	0.042	−0.048
Gene	*RBMS3-AS3*	*HOXA7*	*CTNNA2*	*FBXL19*	*LINC00475*	*SYT8*
coefficient	−0.065	−0.056	0.047	−0.082	−0.072	0.080
Gene	*CYP2S1*	*CDKL3*	*ALX4*	*GPRIN1*	*TRAV16*	*CASP12*
coefficient	0.084	0.053	0.069	0.051	0.044	−0.057
Gene	*TRPC4AP*	*FSTL3*	*TOP3B*			
coefficient	0.050	0.056	0.050			

## Data Availability

Not applicable.

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
