# Peer review of "Microarray-Based Prediction of Polycythemia after Exposure to High Altitudes"

_genes, 2022, doi:10.3390/genes13071193_

Round 1
Reviewer 1 Report
The authors investigate a clinically relevant issue, namely the possibility of identifying high-altitude polycythemia (HAPC) preliminary through genetic analysis.
However, there are several concerns:
1. It remains unclear from which patients the blood samples were taken. Were they Han or Tibetans? How many samples were examined in total?
2. Did the patients give their consent for data collection? Did the institutional review board of your center approved the study?
3. Is there any evidence that the corresponding genes are activated differently in Tibetans? Are there any comparative studies?
Minor findings:
1. Lines 48-53 already include a summary of the results. This section does not need to be in the introduction.
2. Text is missing in section "6. Patents" (Line 292)
Author Response
Thanks for your careful review of our manuscript and your friendly evaluation and affirmation of the value of our work (Manuscript ID genes-1752435, Microarray-based prediction of polycythemia after exposure to high altitudes). All comments and suggestions are very helpful for us. Based on these comments and suggestions, we have revised the manuscript, and the details are as follows. To ensure a complete English revision, we have checked the manuscript in detail and used an advised English editing service.
Thank you again and best wishes.

Reviewer 2 Report
High-altitude polycythemia (HAPC) or commonly known as mountain sickness is a debilitating health issue in individuals that are exposed to hypoxia. Since there is no currently effective treatment it makes sense that disease prevention is the critical strategy towards combating HAPC. Consequently, the aim of this study is to develop a prediction model to discover genes that may be able to predict the development of HAPC on exposure to hypoxia. The authors use several different microarray data sets from individuals exposed to hypoxia and individuals that developed HAPC to uncover genes associated with HAPC development.
1. Can the authors explain how gene expression studies in hypoxia would be able to predict the development of HAPC when the genes of interest are studied at ground level?
2. Can the authors give details about the origins of the two data sets in terms of the age, ethnicity and gender of the participants
3. Data set GSE135109 – did any of the participants develop HAPC at Day 3 of the study?
4. The study identified two genes as possible predictors of HAPC
a. Have their differential expression been profiled over a hypoxia exposure time course
b. Have them been validated in an independent cohort?
c. Please give details of what diseases HCAR3 is a therapeutic target and how this is related to HAPC?
5. On Page 8, lines 209 and 210 the authors state that used “own dataset” to cross validate as reported in supplementary figures.
a. Please clarity to the origins of this “own dataset”
b. No supplementary figures were available with the manuscript
6. The authors state in their Discussions that data from a proteome profiling of HAPC uncovered different series of genes but not the same as their study. They dismissed this as “the biological samples used in the two studies are inconsistent”. Can the authors explain why this is the case.
7. Analysis of data set GSE29977 would reveal any differential expression related to the condition of HAPC and potentially insight into therapeutic options. This was not discussed in the paper.
8. The authors need to clarify the aim of the study as to whether they are profiling the genes involved in development of HAPC or predicting the susceptibility to HAPC.
9. If it is the later aim then they need to consider a different approach that would be based on genetic variation and gene association studies as opposed to hypoxia –induced gene profiling.
1 Abstract line 13 and Introduction line 39. Please clarify if is “between 5 and 18%” or “between 5% and 8%” is correct?
I Introduction line 33 states the reference “Vault in 1980” as number 5 but References line 327 state number 5 as Winslow RM et al in the Journal of Applied Physiology. Which is the correct citation?
1 Discussion line 259 reference Wang, Huang & Gao, 2021 is cited in the correct format and absent from the citation list
1Conclusion lines 288 and 289 states “the gene set for heme metabolism is down regulated” does that include HBA1 and HBA2 in cluster 5?
Author Response
Thank you very much for your careful review of our manuscript and friendly evaluation of our work. All your comments are very helpful for us to improve the manuscript and have also given us great encouragement to continue the work. According to your comments, we have revised the manuscript and itemized responses as follows.

Round 2
Reviewer 2 Report
Many thanks for your email and the revised paper. I have reviewed it and believe it has improved significantly to warrant publication in Genes. However would suggest that the paper would benefit from English language editing.